# Prognostic Relevance of Nuclear Receptors in Relation to Peritumoral Inflammation and Tumor Infiltration by Lymphocytes in Breast Cancer

**DOI:** 10.3390/cancers14194561

**Published:** 2022-09-20

**Authors:** Melitta B. Köpke, Marie-Christine Chateau, Florence Boissière-Michot, Mariella Schneider, Fabian Garrido, Alaleh Zati-Zehni, Theresa Vilsmaier, Mirjana Kessler, Nina Ditsch, Vincent Cavaillès, Udo Jeschke

**Affiliations:** 1Department of Obstetrics & Gynecology, University Hospital Augsburg, 86156 Augsburg, Germany; 2Translational Research Unit, Montpellier Cancer Institute Val d’Aurelle, 208 rue des Apothicaires, F-34298 Montpellier, France; 3Department of Obstetrics and Gynecology, University Hospital, LMU Munich, 81377 Munich, Germany; 4IRCM-Institut de Recherche en Cancérologie de Montpellier, INSERM U1194, Université Montpellier, Parc Euromédecine, 208 rue des Apothicaires, F-34298 Montpellier, France

**Keywords:** tumor-infiltrating lymphocytes (TILs), peritumoral inflammation, nuclear receptors, prognosis, breast cancer

## Abstract

**Simple Summary:**

The aim of this study was to investigate the prognostic impact of tumor-infiltrating lymphocytes (TILs) in a panel of 264 sporadic breast cancers by quantifying TIL levels according to Salgado and correlate this with type I and II nuclear receptor expression. Breast cancer cases with a TIL Salgado score of >15% showed a significantly decreased overall survival and peritumoral inflammation (according to Klintrup) determined the prognostic value of ER, PR, and PPARγ in BC. Therefore, the present study demonstrates significant relations between TIL levels, nuclear receptor expression and prognosis in breast cancer.

**Abstract:**

The prognostic impact of tumor-infiltrating lymphocytes (TILs) is intensively investigated in breast cancer (BC). It is already known that triple-negative breast cancer (TNBC), the most aggressive type of BC, has the highest percentage of TILs. In addition, there is an influence of steroid hormone receptor expression (type I nuclear receptors) on TIL subpopulations in breast cancer tissue. The link between type II nuclear receptors and the level of TILs is unclear. Therefore, the aim of this study was to quantify TILs in a panel of 264 sporadic breast cancers and investigate the correlation of TIL levels with type I and II nuclear receptors expression. TIL levels were significantly increased in the subgroup of TNBC. By contrast, they decreased in estrogen (ER)- or progesterone receptor (PR)-positive cases. Moreover, TIL levels were correlated with type II nuclear receptors, including PPARγ, with a significant inverse correlation of the nuclear form (r = −0.727, *p* < 0.001) and a weak positive correlation of the cytoplasmic form (r = 0.202, *p* < 0.002). Surprisingly, BC cases with a TIL Salgado score of >15% showed a significantly decreased overall survival. In addition, peritumoral inflammation was also quantified in BC tissue samples. In our cohort, although the level of peritumoral inflammation was not correlated with OS, it determined the prognostic value of ER, PR, and PPARγ in BC. Altogether, the present study provides a differentiated overview of the relations between nuclear receptor expression, TIL levels, peritumoral inflammation, and prognosis in BC.

## 1. Introduction

Although the prognostic impact of tumor-infiltrating lymphocyte (TIL) populations in breast cancer (BC) is still debated, triple-negative breast cancer (TNBC) clearly shows a higher density of TILs as compared with other BC subtypes [1,2,3,4]. This is probably due to their higher tumor mutational burden leading to an increased number of antigenic tumor variants and neoepitope load [5]. TILs comprise different populations of lymphocyte subtypes (T and B cells) and natural killer (NK) cells. Furthermore, macrophages and dendritic cells (DCs) are also present in the tumor environment [6,7,8,9,10,11,12]. Originally, peritumoral inflammation was evaluated by a quantification score developed by Klintrup et al. [13] for colorectal cancer, and the inflammatory reaction was divided into four categories. Later on, the morphological evaluation of TILs was assessed by examination of hematoxylin- and eosin-stained tumor sections and standardized by an international group of pathologists, published by Salgado et al. [14] and generally known as the Salgado score [15].

As already stated, TNBCs show a higher density of TILs than other BC subtypes, and estrogen (ER) and progesterone receptor (PR) (type I nuclear receptors) correlation with TIL density has already been described by a number of studies [16,17,18]. In TNBC, on the other hand, stromal TILs are considered a strong prognostic factor, and patients with a high TIL density show better survival [19]. Although the links between TILs and steroid hormone receptors (type I nuclear receptors) are intensely studied in BC [20], the association between type II nuclear receptors and TILs has not been investigated. Type II nuclear receptors form heterodimers with RXR and consist of a variety of subtypes, including thyroid hormone and vitamin D receptors, PPARs, AhR, LXR, and others [21,22,23,24,25,26,27,28]. Their role in BC biology and their impact on patient survival have been reported by our group and others [24,29,30,31,32,33].

Because studies on the link between type II nuclear receptors, TILs, and inflammatory cell reaction in BC are lacking, the aim of this work was to investigate correlations between TILs or peritumoral inflammation and type I and II nuclear receptors and their influence on patient survival.

## 2. Materials and Methods

### 2.1. Materials

This study is based on the use of a cohort consisting of 264 formalin-fixed paraffin-embedded primary BC tissues (see Appendix A) that were collected from patients who underwent surgery between 2000 and 2002 at the Department of Gynecology and Obstetrics of the Ludwig-Maximilian-University in Munich, Germany (clinicopathological characteristics of the patients are provided in the Appendix A). After an observation period of more than 10 years, disease-free survival (DFS) and overall survival (OS) were statistically analyzed. The follow-up data for this cohort were retrieved from the Munich Cancer Registry. The tissue samples used in this study were leftover material after all diagnostics had been completed and were retrieved from the archive of Gynecology and Obstetrics, Ludwig-Maximilian-University, Munich, Germany.

### 2.2. Ethical Approval

All patient data and clinical information from the Munich Cancer Registry were fully anonymized and encoded for statistical analysis. The study was performed according to the standards set in the Declaration of Helsinki 1975. The current study was approved by the ethics committee of the Ludwig-Maximilian-University Munich, Germany (approval number 048–08). The authors were blinded from the clinical information during experimental analysis.

### 2.3. Expression of Nuclear Receptors

Using the above-described BC cohort, the expression of type I and type II nuclear receptors has been previously analyzed by immunohistochemistry by our group. Information was specifically evaluated regarding ERα and PR [34], PPARγ [35], thyroid hormone receptors (TRs) [36,37], AhR [33], LXR [38], and RXRα [39,40].

### 2.4. TIL Quantification

Tumor-infiltrating lymphocytes (TILs) were quantified by an experienced gynaeco-pathologist (M-C.C). Figure 1 shows representative pictures of low and high expression of the evaluated nuclear receptors. Scoring was based on the method developed by Salgado et al. specifically for the evaluation of TILs in BC tissue [14]. According to this method, stromal TILs within the tumor are scored as a percentage of the stromal areas alone (areas occupied by carcinoma cells are not included in the assessed area).

We also adapted the scoring method of Klintrup et al. [13], developed originally for the quantification of inflammatory cell reaction in colorectal cancer at the invasive margin, therefore representing immune cells around the tumor, and classified into four categories:score 0 = no inflammatory cells at the invasive margin;score 1 = mild and patchy increase of inflammatory cells at the invasive margin;score 2 = increased inflammatory cells forming a band-like infiltration at the invasive margin;score 3 = prominent inflammatory reaction forming a cup-like zone at the invasive margin.

### 2.5. Statistical Evaluation

Statistical analysis was performed with the IBM Statistical Package for the Social Sciences (IBM SPSS Statistic v26.0 Inc., Chicago, IL, USA). The gathered results were inserted into the SPSS database in the implied manner. Correlations between findings of immunohistochemical staining were performed with Spearman’s analysis. The nonparametric Kruskal–Wallis for more than two independent groups or the Mann–Whitney U test was used to test for differences in TIL density regarding the set prognostic markers. OS (in years) and DFS (in years) were compared by Kaplan–Meier graphics, and differences in patient survival times were tested for significance using the chi-square statistics of the log-rank test. For multivariate analyses, the Cox regression model for survival was used, and the following factors were included: age of the patient, pT and pN of the TNM staging system, grading, and histology type. Each parameter considered significant showed a value of *p* < 0.05. The *p*-value and the number of patients analyzed in each group are given for each chart.

## 3. Results

### 3.1. Quantification of TILs in the Tumor Stroma

Quantification of TILs according to the Salgado score revealed that the majority of the tumors exhibited no more than 10% TILs, with 183 cases (69,3%) showing up to 10% TILs, whereas only 26 cases (9.9%) showed more than 40% TILs (exact distribution is presented in Table 1).

### 3.2. TIL Density According to BC Subtypes

As expected, TNBC cases showed the highest TIL counts (13.33% of all cases), as evaluated by the Salgado score (median 26.6%; triple negative and 6.0%; remaining cases; *p* < 0.001; Figure 2A). In addition, TIL density was significantly elevated (*p* = 0.008) from G1 to G3 carcinomas (Figure 2B). Grading was performed according to the Elston and Ellis criteria [41]. Comparing the BC molecular subtypes, we found significantly higher TIL levels in basal-like and in the two Her2 subtypes (luminal and nonluminal) in comparison with luminal A and luminal B molecular subtypes (Figure 3). Appendix A contains patient numbers for all groups.

### 3.3. Prognostic Relevance of TIL Levels

Kaplan–Meier curve visualized a significant negative association of the OS (Figure 4) when TIL levels were higher than 15%, as assessed by the Salgado score in the whole BC population. A statistically negative significant correlation was observed for the OS (*p* = 0.02), calculated by the log-rank test?

However, as shown in Table 2, multivariate Cox regression did not identify the level of TILs as an independent prognostic factor for OS (HR 1.967, 95%CI 0.921–4.200, *p* = 0.081). Only age at surgery was significant.

### 3.4. Correlation of TILs with Nuclear Receptor Expression

As expected, BC cases with ER expression (80% of all cases) showed a significantly lower level of TILs as evaluated by the Salgado score (median 9.3%; ERα positive and 23.3%; ERα negative; *p* < 0.001; Figure 5A). A similar result was obtained by analyzing PR expression; PR-positive BC (57% of all cases) also showed a significantly lower expression of TILs as evaluated by the Salgado score (median 15.7%; PR positive and 16.9%; PR negative; *p* = 0.003; Figure 5B).

Significant correlations were also observed with type II nuclear receptors. Indeed, the nuclear forms of PPARγ and LXR showed a negative correlation with TIL density. By contrast, the level of TRβ and RXRα expressed in the nucleus showed a very weak positive correlation with TIL density. In addition to the nuclear expression, type II nuclear receptors were also detected in the cytoplasm. We identified weak but significant positive correlations between TILs and cytoplasmic type II receptors (Table 3).

### 3.5. Quantification of Peritumoral Inflammation

Quantification of peritumoral inflammation adapted from the Klintrup score revealed 74 cases (30%) with no inflammatory cells at the invasive margin, 143 cases (57.9%) with mild and patchy increase in inflammatory cells, and 30 cases (12.1%) with increased inflammatory cells forming a band-like infiltration at the invasive margin (Table 4). None of the 247 assessable samples showed a prominent inflammatory reaction at the invasive margin (score 3, according to Klintrup criteria). A total of 17 cases were not assessable. Examples of tumors with different TIL densities according to the Klintrup score is shown in Figure 6. 

As shown in Figure 7, quantification of peritumoral inflammation on BC tissue samples according to the Klintrup score was not associated with OS differences.

### 3.6. Correlation of Peritumoral Inflammation with Nuclear Receptor Expression

Concerning nuclear receptor expression and peritumoral infiltration, we found significantly more ER- and PR-positive cases in patients with no or mild and patchy increase in inflammatory cells at the invasive margin compared with cases with increased inflammatory cells forming a band-like infiltration at the invasive margin (*p* < 0.001 and 0.017, respectively).

Analyses of type II receptors revealed a significant increase in cytoplasmic PPARγ in cases with no or mild and patchy increase in inflammatory cells compared with cases with increased inflammatory cells (Figure 8A, *p* = 0.002). By contrast, expression of PPARγ in the nucleus was significantly reduced in cases with no or mild and patchy increase in inflammatory cells compared with cases with increased inflammatory cells (Figure 8B, *p* = 0.003). Analyses of other type II receptors also revealed a significant correlation between cytoplasmic THRβ or nuclear LXR and peritumoral inflammation (Suppl. Figures, *p* = 0.045 and 0.026, respectively).

Type-specific analyses (luminal A, luminal B, basal-like, HER2-positive) were performed to correlate peritumoral inflammation with type II nuclear receptors but did not yield significant results due to the small number of cases (data not shown).

### 3.7. Prognostic Relevance of Nuclear Receptors Expression according to Peritumoral Inflammation 

Although peritumoral inflammation had no prognostic relevance in our BC cohort, we asked whether it might influence the prognostic relevance of nuclear receptors, including ERα and PR. ERα has been known for decades as a positive prognostic marker for BC survival. We analyzed OS of Erα-positive and -negative patients according to peritumoral inflammation. Kaplan–Meier survival analyses visualized a significant positive correlation of ERα expression only in the subgroups with no (score 0) inflammatory cells at the invasive margin. Patients being ERα-negative showed a mean survival time of 4.2 years compared with 10.3 years for ERα-positive patients (Figure 9A, *p* < 0.001). There is also a significant difference in OS between ERα-positive and -negative patients in the tumor with score 1, i.e., those with mild and patchy increase in inflammatory cells at the invasive margin. Patients being ERα-negative showed a mean survival time of 8.0 years compared with 10.6 years for Erα-positive patients (Figure 9B, *p* = 0.013). By contrast, in patients with an increased inflammatory pattern at the invasive margin (score 2), the lack of ERα expression was no longer a marker of poor prognosis (Figure 9C, *p* = 0.642).

Progesterone receptor is also a positive prognostic marker for BC survival. We analyzed OS of PR-positive and -negative patients according to peritumoral inflammation. Kaplan–Meier survival analyses visualized a significant positive association with long OS if PR is positive only in the subgroup with no inflammatory cells at the invasive margin (Figure 10A). Patients being PR-positive showed a mean survival time of 10.6 years compared with 7.6 years for PR-negative patients (*p* = 0.033). By contrast, there was no significant difference in OS between PR-positive and -negative patients having tumors with mild or increased peritumor inflammation (Figure 10B,C, *p* = 0.070 and *p* = 0.319, respectively).

In contrast to type I nuclear receptors, cytoplasmic PPARγ type II nuclear receptor was recently reported by our group to be a negative prognostic marker for breast cancer survival [35]. We analyzed the overall survival of cytoplasmic PPARγ-positive and -negative patients in the three above-described subgroups (score 0–2 of peritumoral inflammation). Kaplan–Meier survival analysis visualized a significant negative association of cytoplasmic PPARγ with OS only in the subgroup with no inflammatory cells at the invasive margin (Figure 11A). Patients with tumors being negative for cytoplasmic PPARγ showed a mean survival time of 10.7 years compared with 7.7 years for patients with tumors positive for cytoplasmic PPARγ (*p* = 0.008). No significant difference in OS was observed between cytoplasmic PPARγ-positive and -negative patients in BC with mild or increased peritumoral inflammation (Figure 11B,C, *p* = 0.280 and C, *p* = 0.225, respectively). As shown in Appendix A, no significant differences were observed concerning the correlation of RXRα, LXR, and AhR expression with patient survival in relation to the levels of peritumoral inflammation. Finally, when the same type of analyses as the ones shown in Figure 9, Figure 10 and Figure 11 was performed using a classification based on the Salgado score, we observed no influence on the correlation between ERα, PR, or PPARγ expression and survival (data not shown). This could be explained by the fact that the two quantification scores exhibited differences in tumor classification, as shown in Appendix A. Although all tumors with high peritumoral inflammation had a TIL density of >30%, the groups with no or mild peritumoral inflammation were more heterogeneous in terms of TIL levels.

## 4. Discussion

Within this study, we combined the analysis of TIL levels (following Salgado’s guidelines), peritumoral inflammation (adapted from Klintrup criteria), and nuclear receptor expression in a cohort of BC in relation to patient overall survival (OS). Taking into account both nuclear and cytoplasmic expression of type I and II nuclear receptors, as former studies already showed a prognostic impact of nuclear receptor subcellular localization [40], our results highlight a strong interplay of the two parameters, which determines their prognosis value for BC patients. Next to the influence of TILs on tumor progression and prognosis, the nuclear receptors and their subcellular localization play notable roles in the pathophysiology of BC as well [34,42]. As the subcellular localization of nuclear receptors, e.g., RXRα or PPARγ, have such an impact on prognosis, one can assume that they exert specific functions in the cytoplasm, as has been proposed for PPARs [43].

In concordance with already published results, we found significantly higher rates of TILs in triple-negative breast cancer (TNBC). As known from former studies, TNBCs showed a higher density of TILs than other BC subtypes, probably because of their higher number of antigenic tumor variants, neoepitope load, and tumor mutational burden [5]. In TNBC, stromal TILs are considered a strong prognostic factor, and patients with a high TIL density show better survival [1,2,3,44,45,46,47,48,49]. However, in our panel, no effect of TILS on clinical outcome was observed in the group of TNBC, probably because the number of samples was too small (34 cases) to reach significance. In addition, we confirmed that TIL levels are significantly elevated in Her2-positive and basal-like breast cancer [50]. Luminal types of BC showed the significantly lowest level of TILs in our study group.

As expected, in our BC cohort, ERα- or PR-negative cases showed higher TIL density. Interestingly, cases with high cytoplasmic PPARγ expression also showed significantly elevated TIL levels. In a recent study on the same patient cohort, we analyzed the combined cytoplasmic expression of RXRα and PPARγ. Patients with tumors expressing both NRs in the cytoplasm of tumor cells exhibited significantly shorter OS and DFS [39]. Based on those results, we investigated the correlation between TIL levels and the expression of nuclear type II receptors. The main member of this group is RXRα because all other members of this subfamily form heterodimers with RXRα. Interestingly, nuclear RXRα showed a positive correlation with TIL density as only the thyroid hormone receptor TRβ did, whether the latter was expressed in the nucleus or the cytoplasm of tumor cells. All other type II receptors (PPARγ, TRα, TRα1, and TRβ) showed a positive correlation with TIL density only if expressed in the cytoplasm. Nuclear expression of PPARγ and LXR resulted in a negative correlation with TIL levels. To our knowledge, the concordance of TIL density and type II nuclear receptor expression was not investigated before.

Concerning the correlation between TILs and survival, we could show that TILs (using the Salgado score and 15% TILs as cutoff value) have poor prognostic value in OS in our BC cohort. This result might be explained by the fact that the studied cohort is highly heterogeneous in BC subtypes, ER, and HER2 status or histology (ductal, lobular, and others). In this cohort, and as expected, the basal samples (a subpopulation that includes TNBC in a vast majority) are those displaying the highest TIL density. It is well known that these tumors are also those with the worst clinical outcome. Indeed, whereas it is well known that TILs are associated with better clinical outcomes in TNBC- and HER-2 positive breast cancer [51,52], the study of Desmedt, Salgado et al. described that TIL levels were associated with worse prognostic outcomes in lobular carcinoma [50]. Thus, the association of poor clinical outcomes with high TILs density could reflect sample heterogeneity.

We then analyzed the influence of the peritumoral infiltrate on the prognostic value of nuclear receptors. Although the prognostic value of TIL levels in the ER-negative/Her2-negative breast cancer population is well known, the impact of immune infiltration on the prognosis value of NRs was never investigated before [53,54,55]. ERα (and steroid hormone receptor expression in general) is known to be favorable concerning OS [56,57,58] in BC. Very interestingly, our study clearly showed that the positive correlation of steroid hormone receptor expression with prognosis is dependent on the level of peritumoral inflammation. Indeed, ERα exhibited a prognosis value only for patients with tumors having low levels of inflammation at the invasive margin. A similar result was obtained with the progesterone receptor (PR): only the subgroup with no peritumoral inflammation showed significant differences in overall survival based on PR positivity. In concordance with the results obtained with the steroid hormone receptor, the cytoplasmic expression of PPARγ was found to be a negative prognosticator only in the group without peritumoral inflammation.

Concerning the impact of the immune peritumoral infiltrate on the prognosis value of ERα, the results shown in Figure 9 confirmed that some ER-negative tumors are not immunogenic (at least do not present an inflammatory cell reaction) and, consequently, are more aggressive and lead to a short OS of patients. The molecular mechanisms of this observation remain to be deciphered but could be related to Ras/MAPK pathway activation in these TNBC samples. Indeed, it has been reported that this pathway may promote host antitumor immune evasion in tumor cell-autonomous pathways [59]. A striking result was the link between cytoplasmic expression of PPARγ and peri- or intratumor immune infiltration. Cytoplasmic expression of PPARγ was accompanied with an increase in immune infiltration. It is already known that the expression of PPARγ, as the key regulator of lipogenesis, is altered in breast cancer [60]. We could show very recently that cytoplasmic PPARγ is a negative prognosticator in breast cancer [35,39]. PPARγ can determine the cellular phenotype by regulating differentiation and function by activating the transcription of PPARγ target genes [61]. Similar molecular mechanisms are in place in immune cells, and also, here, PPARγ can determine cellular phenotype [61].

This study has some limitations, considering its retrospective nature, the relative heterogeneity of the cohort, and the way TILs was assessed. For instance, it might have been interesting to analyze the influence of peritumoral infiltrate on the prognostic value of nuclear receptors within different molecular subgroups of BC, in particular in the TNBC. In addition, we herein only performed a global analysis of TILs, and since these cells may be immunogenic or immune-suppressive, more precise methods based on immunohistochemical detection of the different lymphocyte subtypes (including cytotoxic and regulatory T cells, or B/plasma cells) would have been more informative. These points will be addressed in further studies, which are also needed to determine whether the prognosis value of cytoplasmic PPARγ, combined with a lack of peritumoral infiltration, could become important in clinical routine and influence therapy decisions.

## 5. Conclusions

Altogether, this study is one of the first that describes the correlation between the expression of nuclear receptors (in particular PPARγ) and peritumoral inflammation or TIL levels in relation to prognosis in BC. Although some studies exist on PPARα in melanoma [62,63,64], PPARγ has, to our knowledge, never been investigated in relation to TILs and survival in any form of cancer. In our whole cohort of sporadic breast cancers, PPARγ expression in the cytoplasm of cancer cells was positively correlated with TIL levels and peritumoral inflammation. In addition, the prognostic value of cytoplasmic PPARγ was determined by the level of immune peritumoral infiltrate.

## Figures and Tables

**Figure 1 cancers-14-04561-f001:**
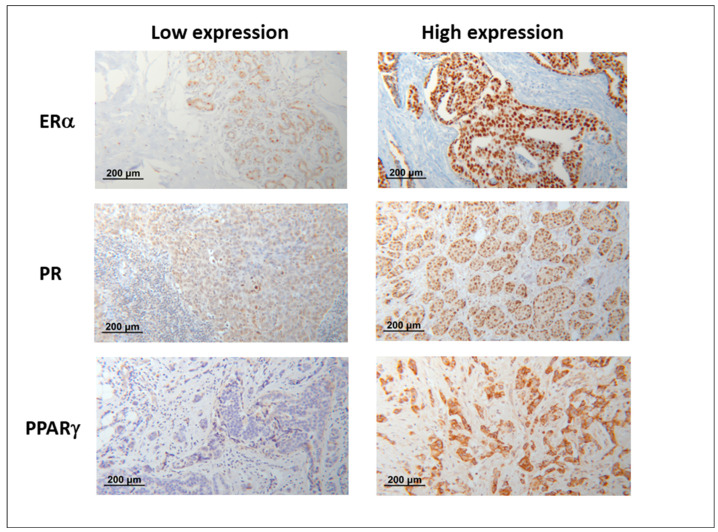
Representative microphotographs of low and high expression of ERα (estrogen receptor alpha), PR (progesterone receptor), and PPARγ (peroxisome proliferator-activated receptor gamma) in 25× magnification.

**Figure 2 cancers-14-04561-f002:**
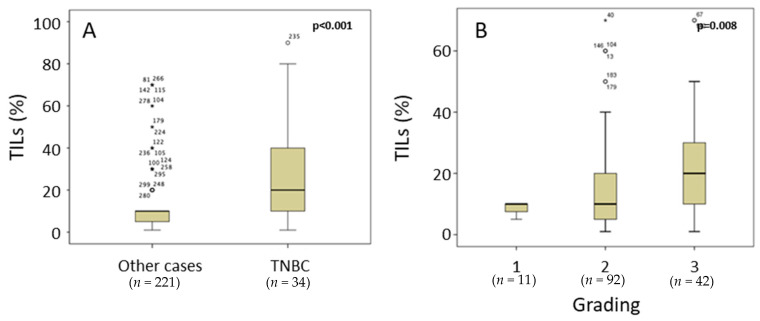
TIL density, according to Salgado, is significantly higher in cases with triple-negative tumors compared with remaining cases ((**A**); *p* < 0.001) and increases significantly from G1 to G3 tumors ((**B**), *p* = 0.008). Mild outliers are marked with a circle (O) on the boxplot. Extreme outliers are marked with an (★) on the boxplot.

**Figure 3 cancers-14-04561-f003:**
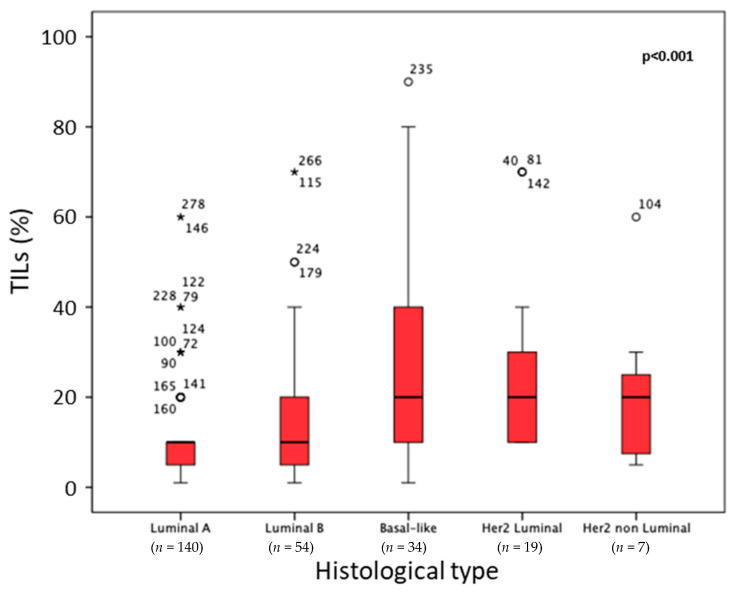
TIL density is significantly higher in basal-like- and Her2-positive cases compared with luminal A and luminal B cases (*p* < 0.001). The numbers represent outliers, and the circles represent outlier cases. Extreme outliers are marked with an (★) on the boxplot.

**Figure 4 cancers-14-04561-f004:**
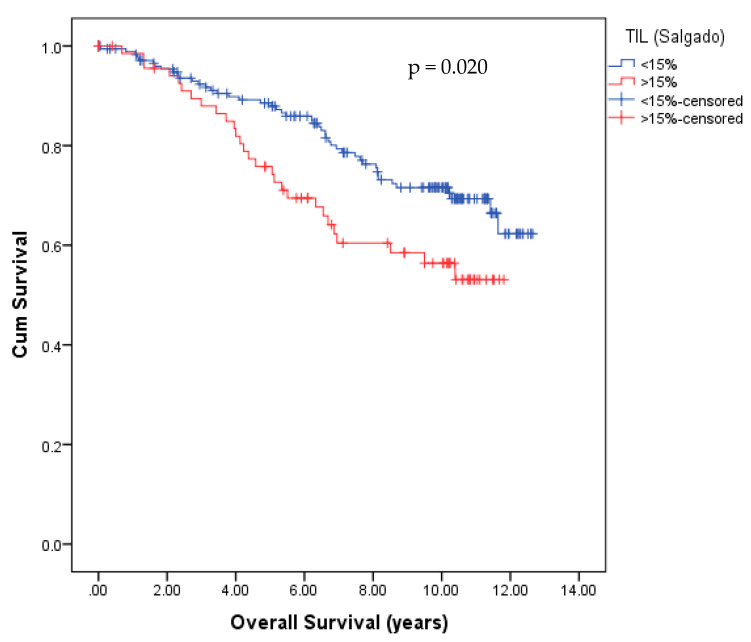
Kaplan–Meier survival analyses of TIL density, according to the Salgado score, revealed significant differences in OS. Patients with TIL levels greater than 15% showed significantly reduced OS (mean 8.6 years) compared with patients with lower TIL levels (mean OS 10.4 years; *p* = 0.020).

**Figure 5 cancers-14-04561-f005:**
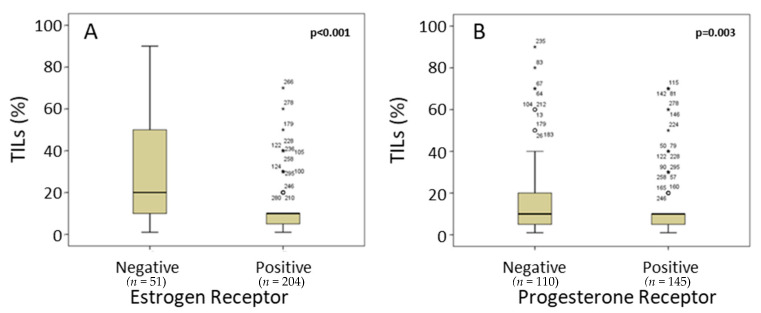
Level of TILs evaluated by the Salgado score is significantly higher in cases with no expression of ERα (compared with cases with ERα expression ((**A**); *p* < 0.001). Level of TILs is also significantly higher in cases with no PR expression ((**B**); *p* = 0.003). The numbers represent outliers, and the circles represent outlier cases. Extreme outliers are marked with an (★) on the boxplot.

**Figure 6 cancers-14-04561-f006:**
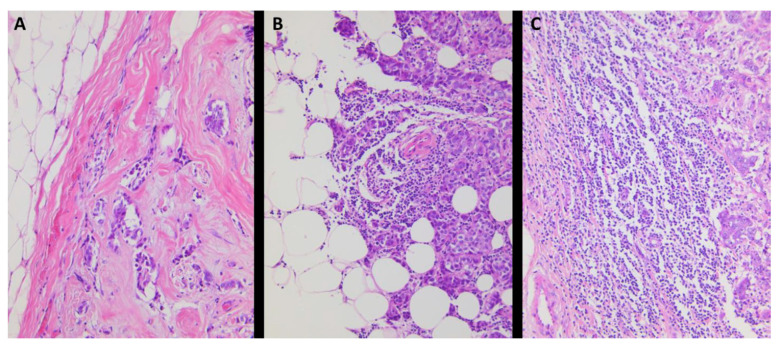
Examples of tumors with different levels of peritumoral inflammation according to the Klintrup score. (Panel (**A**): score 0; panel (**B**): score 1; Panel (**C**): score 2); all pictures 20× lens magnification.

**Figure 7 cancers-14-04561-f007:**
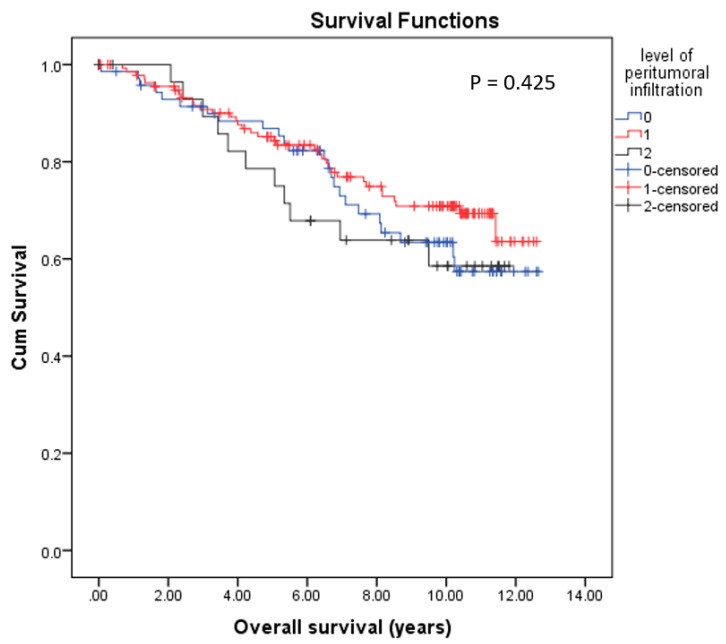
Kaplan–Meier survival analyses of the level of peritumoral inflammation in relation to overall survival (OS) showed no significant differences (*p* = 0.425).

**Figure 8 cancers-14-04561-f008:**
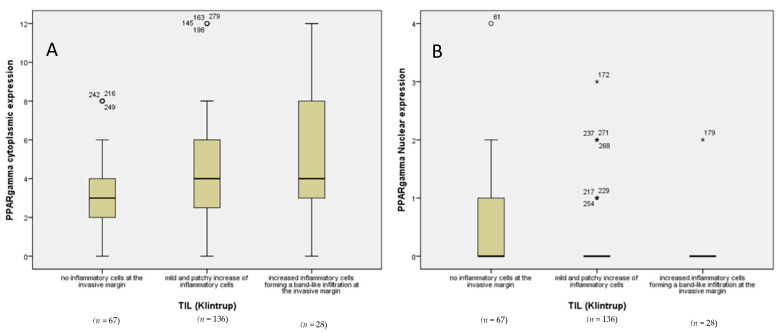
Expression of PPARγ varies according to peritumoral inflammation. Cytoplasmic PPARγ is significantly higher in cases with peritumoral immune infiltration (**A**), whereas an inverse correlation is observed with the levels of nuclear PPARγ (**B**). Mild outliers are marked with a circle (○) on the boxplot. Extreme outliers are marked with an (★) on the boxplot.

**Figure 9 cancers-14-04561-f009:**
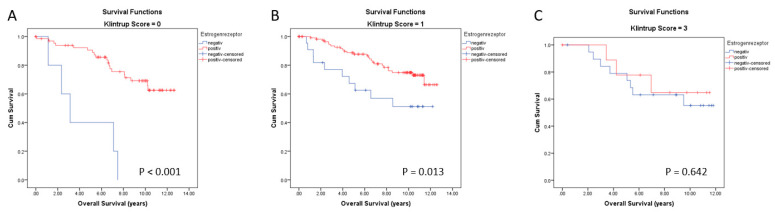
Kaplan–Meier survival analyses for estrogen receptor (ERα) according to the level of peritumoral inflammation with no inflammatory cells at the invasive margin (Klintrup Score 0 (**A**)), mild and pathy increase of inflammatory cells (Klintrup Score 1(**B**)) and increased inflammatory cells (Klintrup Score 2 (**C**)) on overall survival (OS).

**Figure 10 cancers-14-04561-f010:**
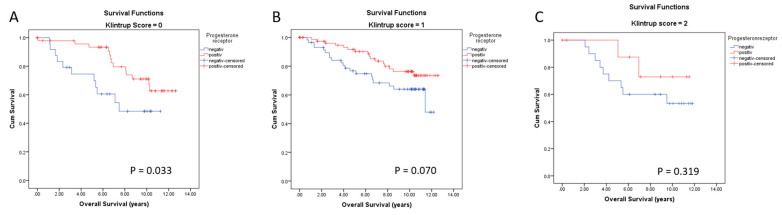
Kaplan–Meier survival analyses for progesterone receptor (PR) in relation to the level of peritumoral inflammation with no inflammatory cells at the invasive margin (Klintrup Score 0 (**A**)), mild and pathy increase of inflammatory cells (Klintrup Score 1(**B**)) and increased inflammatory cells (Klintrup Score 2 (**C**)) on overall survival (OS).

**Figure 11 cancers-14-04561-f011:**
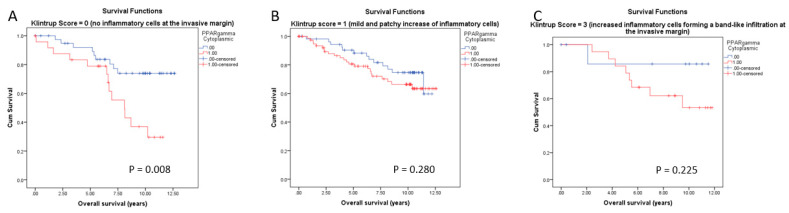
Kaplan–Meier survival analyses for cytoplasmic PPARγ in relation to the level of peritumoral inflammation with no inflammatory cells at the invasive margin (Klintrup Score 0 (**A**)), mild and pathy increase of inflammatory cells (Klintrup Score 1(**B**)) and increased inflammatory cells (Klintrup Score 2 (**C**)) on overall survival (OS).

**Table 1 cancers-14-04561-t001:** TIL quantification according to the Salgado score.

% TILs	Frequency	Percent	Cumulative Percent of Accessible Cases
1	32	12.1	12.5
5	56	21.3	34.5
10	95	35.9	71.8
20	27	10.2	82.4
30	19	7.2	89.8
40	8	3.0	92.9
50	4	1.5	94.5
60	5	1.9	96.5
70	7	2.7	99.2
80	1	0.4	99.6
90	1	0.4	100
Not assessable	9	3.4	
Total	264	100	

**Table 2 cancers-14-04561-t002:** Multivariate Cox regression analysis of TIL levels (Salgado score) regarding OS.

	Significance	Hazard Ratio	95% CI	
			Lower	Upper
TIL level (>15%)	0.081	1.967	0.921	4.200
Age at surgery	0.005	1.037	1.011	1.064
pT	0.153	1.193	0.936	1.520
pN	0.220	1.167	0.912	1.492
Stage	0.527	0.694	0.224	2.153
Grading	0.107	1.683	0.893	3.171
Histology	0.460	1.010	0.984	1.036

**Table 3 cancers-14-04561-t003:** Correlation of nuclear and cytoplasmic staining of type II nuclear receptors with TILs assessed by the Salgado score (green or orange: weak negative or positive correlation [r] < 0.39; red: strong negative correlation [r] > 0.60).

Nuclear Receptor	Parameter	Value
	N	255
Nuclear PPARγ	Correlation Coefficient	−0.727
	Sig. (2-tailed)	<0.001
	N	237
Cytoplasmic PPARγ	Correlation Coefficient	0.202
	Sig. (2-tailed)	0.002
	N	237
Nuclear LXR	Correlation Coefficient	−0.254
	Sig. (2-tailed)	<0.001
	N	250
Cytoplasmic TRα	Correlation Coefficient	0.197
	Sig. (2-tailed)	0.002
	N	240
Cytoplasmic TRα1	Correlation Coefficient	0.203
	Sig. (2-tailed)	0.002
	N	237
Nuclear TRβ	Correlation Coefficient	0.172
	Sig. (2-tailed)	0.007
	N	242
Cytoplasmic TRβ	Correlation Coefficient	0.279
	Sig. (2-tailed)	<0.001
	N	242
Nuclear RXRα	Correlation Coefficient	0.137
	Sig. (2-tailed)	0.029
	N	255

**Table 4 cancers-14-04561-t004:** Quantification of peritumoral inflammation according to the Klintrup score.

Peritumoral Inflammation	Frequency	Percent	Cumulative Percent of Accessible Cases
Score 0—no inflammatory cells at the invasive margin	74	28.0	30
Score 1—mild and patchy increase of inflammatory cells at the invasive margin	143	54.2	87.9
Score 2—increased inflammatory cells forming a band-like infiltration at the invasive margin	30	11.4	100
Not assessable	17	6.4	
Total	264	100	

## Data Availability

Data supporting the reported results can be obtained from the corresponding author.

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
