# Peer review of "Prognostic Relevance of Nuclear Receptors in Relation to Peritumoral Inflammation and Tumor Infiltration by Lymphocytes in Breast Cancer"

_cancers, 2022, doi:10.3390/cancers14194561_

Round 1
Reviewer 1 Report
Kopke et al. present their results about tumor infiltrating lymphocytes in sporadic breast cancer and their correlation to type I and II nuclear receptors expression. Thereby, Kopke initially differentiates the cohort based on breast cancer types and the stromal TIL level before accessing intra-tumoral nuclear receptor expression in all those.
The most obvious point that needs addressing is the setup and way TIL involvement is evaluated. The authors only stain for all while it is quite commonly known that different types of TIL (cytotoxic T cells, vs. regulatory T or B/plasma cells) have different implications within the immune tumor microenvironment. Some of those effects might be immunogenic or immune suppressive and therefore should be evaluated separately to get conclusive results.
It is quite confusing why the receptor expression of the tumor cells are considered and not the TIL infiltration within the tumor. The true immunogenic potential should not only be assessed only based on stromal and marginal infiltration as the immune systems ability to invade throughout the tumor is more telling and informative in regards to survival.
The manuscript would further benefit from the complete multivariate analysis to identify the independent effects of various parameters, e.g. KM analysis in figure 4 could very much be influenced by clinicopathological parameters as age, residual disease status, stages etc. Overall, the analysis of those parameters impact appears incomplete - which only using the T and N values of TNM staging and not incorporating the influence of potential metastasis?
As pointed out in Fig. 2 differences between the types of breast cancer exists in regards to the observable TIL levels. Unfortunately the very same thought is not brought into the analysis of the nuclear receptor expression and all sporadic breast cancers are evaluated in one light. Would those differences not also justify type-specific analysis too, as different conclusion could be drawn based on that?
While the sample numbers are listed in suppl. table 1, the readability would be improved if those numbers were included in the figures respectively.
In conclusion, while presenting some new and interesting data, major revision is recommended to improve the current manuscript.
Author Response
Reviewer 1:
Kopke et al. present their results about tumor infiltrating lymphocytes in sporadic breast cancer and their correlation to type I and II nuclear receptors expression. Thereby, Kopke initially differentiates the cohort based on breast cancer types and the stromal TIL level before accessing intra-tumoral nuclear receptor expression in all those.
- The most obvious point that needs addressing is the setup and way TIL involvement is evaluated. The authors only stain for all while it is quite commonly known that different types of TIL (cytotoxic T cells, vs. regulatory T or B/plasma cells) have different implications within the immune tumor microenvironment. Some of those effects might be immunogenic or immune suppressive and therefore should be evaluated separately to get conclusive results.
The referee is right and we agree that it might have been interesting to quantify the nature of TILs present in the different samples. This might have been done using more precise methods mainly based on immunohistochemistry to detect markers of the different lymphocyte subtypes. However, although some TILs may be immunogenic or immune suppressive, this subdifferentiation is not part of clinical routine nowadays, only the information of the presence of TILs. Therefore, we decided to use the Klintrup- and Salgado-Score as validated scores for clinical routine. Nevertheless, even though the subdifferentiation was not part of this project, it is a goal for our future projects.
- It is quite confusing why the receptor expression of the tumor cells are considered and not the TIL infiltration within the tumor. The true immunogenic potential should not only be assessed only based on stromal and marginal infiltration as the immune systems ability to invade throughout the tumor is more telling and informative in regards to survival.
The aim of this project was to evaluate the prognostic impact of TILs within the tumor (assessed by Salgado-Score, specifically developed for the evaluation of TILs in BC tissue) as well as peritumoral TILs (assessed by adapted Klintrup-Score, developed originally for the quantification of inflammatory cell reaction in colorectal cancer at the invasive margin).
To assess intratumoral TILs, we referred to the Salgado method, which has been largely used by others. The quantification of stromal intratumorally TILs is a good reflect of the immune infiltration. The Klintrup-score on the other hand assesses TILs at the invasive margin, which showed a prognostic value in this cohort.
- The manuscript would further benefit from the complete multivariate analysis to identify the independent effects of various parameters, e.g. KM analysis in figure 4 could very much be influenced by clinicopathological parameters as age, residual disease status, stages etc. Overall, the analysis of those parameters impact appears incomplete - which only using the T and N values of TNM staging and not incorporating the influence of potential metastasis?
In the studied breast cancer panel, there were no cases with metastasis at the time of surgery and material extraction for TIL assessment, therefore metastasis was not included in the multivariate analysis. Moreover, information on R-status after surgery was not available for this panel.
However, we have now augmented the Table 2 with other parameters including stage and histology. As shown in the new Table 2, only age now appears as an independent prognosis factor for OS. Simply with the parameters analyzed in the initial Table 2 (only age, pT, pN, Grading), TILs did not appear as an independent prognosis factor for OS. This observation does not change with the addition of other factors to the multivariate analysis.
- As pointed out in Fig. 2 differences between the types of breast cancer exists in regards to the observable TIL levels. Unfortunately, the very same thought is not brought into the analysis of the nuclear receptor expression and all sporadic breast cancers are evaluated in one light. Would those differences not also justify type-specific analysis too, as different conclusion could be drawn based on that?
We agree with the referee that it might be very interesting to perform the same type of analysis on more homogeneous cohorts corresponding to the different molecular types of breast cancer that have been identified. These type-specific analyses were performed but did not yield significant results due to the small number of cases corresponding to each subtype. Corresponding sentences were included in the result-section (line 232 ff.) as well as in the discussion (line 359 ff.). As the present cohort cannot afford this possibility of type-specific analyses, we are currently setting up new cohorts, in particular for TNBC, to answer this question.
- While the sample numbers are listed in suppl. table 1, the readability would be improved if those numbers were included in the figures respectively.
As suggested by the referee, we have included sample numbers in the Figures 2, 3, 5 and 8.
In conclusion, while presenting some new and interesting data, major revision is recommended to improve the current manuscript.
We thank the referee for all these comments. We have done major revision of the parts suggested by the reviewer and have clearly improved the manuscript. We hope that it will now be acceptable in Cancers.

Reviewer 2 Report
Review of the manuscript untitled : « Prognostic relevance of nuclear receptors in relation to peritumoral inflammation and tumor infiltration by lymphocytes in breast cancer »
In this manuscript the authors searched for the prognosis value of nuclear receptors expression and tumor infiltrating lymphocytes in a big cohort of breast cancer samples.
The manuscript is well written and the results presented are sound. It is of interest to study nuclear receptors not commonly studied. Even if the study is descriptive, it remains of interest for readers.
Comments :
- I suggest to show images of the different staining observed for each nuclear receptor.
- The dual localization of proteins linked with potentially different function should discussed and illustrated with examples in the discussion section.
Author Response
Reviewer 2:
Comments and Suggestions for Authors
Review of the manuscript untitled: « Prognostic relevance of nuclear receptors in relation to peritumoral inflammation and tumor infiltration by lymphocytes in breast cancer »
In this manuscript the authors searched for the prognosis value of nuclear receptors expression and tumor infiltrating lymphocytes in a big cohort of breast cancer samples.
The manuscript is well written and the results presented are sound. It is of interest to study nuclear receptors not commonly studied. Even if the study is descriptive, it remains of interest for readers.
Comments :
- I suggest to show images of the different staining observed for each nuclear receptor.
Corresponding images for low and high staining of ERα, PR and PPARγ have been inserted as Figure 1.
- The dual localization of proteins linked with potentially different function should discussed and illustrated with examples in the discussion section.
In the discussion, the paragraph on subcellular localization of nuclear receptors was expanded for a more detailed discussion of the influence of dual localization of nuclear receptors (line 300 ff.):
“Taking into account both nuclear and cytoplasmic expression of type I and II nuclear receptors, as former studies already showed a prognostic impact of nuclear receptors subcellular localization, our results highlight a strong interplay of the two parameters, which determines their prognosis value for BC patients. Next to the influence of TILs on tumor progression and prognosis, the nuclear receptors and their subcellular localization play notable roles in the pathophysiology of BC, as well. As the subcellular localization of nuclear receptors e.g. RXRα or PPARγ have such an impact on prognosis, one can assume that they exert specific functions in the cytoplasm, as it has been proposed for PPARs.”

Round 2
Reviewer 1 Report
The authors did address all made comments in their response letter. Unfortunately, not all those comments were considerably incorporated in the manuscript, especially regarding the lack of lymphocyte differentiation in their evaluation.
Since Cancers is not a solely clinical focus journal, the response 'something not being done in a clinical routine' is not good enough reasons to not address the point made. I understand that a re-analysis of the samples might not be feasible due to usage of primary tumor tissue. However, the author should at least acknowledge this as a limitation of the study within the discussion section.
In Figure 1 a scale bar would be a better and preferred option over the noted magnification in the text.
Author Response
Reviewer 1 (round 2):
The authors did address all made comments in their response letter. Unfortunately, not all those comments were considerably incorporated in the manuscript, especially regarding the lack of lymphocyte differentiation in their evaluation. Since Cancers is not a solely clinical focus journal, the response 'something not being done in a clinical routine' is not good enough reasons to not address the point made. I understand that a re-analysis of the samples might not be feasible due to usage of primary tumor tissue. However, the author should at least acknowledge this as a limitation of the study within the discussion section.
- We deeply apologize. The referee is right and we should have acknowledged the lack of lymphocyte differentiation in the evaluation as a limitation of the study. We have now added the limitations of the work in the last paragraph of the discussion.
This study has some limitations, considering its retrospective nature, the relative heterogeneity of the cohort and the way TILs was assessed. For instance, it might have been interesting to analyze the influence of peritumoral infiltrate on the prognostic value of nuclear receptors within different molecular subgroups of BC, in particular in the TNBC. Besides, we herein only performed a global analysis of TILs and, since these cells may be immunogenic or immune suppressive, more precise methods based on immunohistochemical detection of the different lymphocyte subtypes (including cytotoxic and regulatory T cells, or B/plasma cells) would have been more informative. These points will be addressed in further studies, which are also needed to determine whether the prognosis value of cytoplasmic PPARg combined to a lack of peritumoral infiltration could become important in clinical routine and influence therapy decisions.
In Figure 1 a scale bar would be a better and preferred option over the noted magnification in the text.
- We have added the scale bar as requested in Figure 1.
